# Diversity of Carbon Storage Economics in Fertile Boreal Spruce (*Picea Abies*) Estates

**Petri P. Kärenlampi** 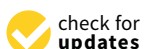

Lehtoi Research, 81235 Lehtoi, Finland; petri.karenlampi@professori.fi

**Abstract:** A "normal forest", an idealized estate with a uniform distribution of stand ages, can be used in the study of sustainable management practices. As the normal forest contains a variety of stand ages, the characteristics of the stands can be represented in terms of a "normal stand", with properties known as a function of age. This paper takes seven never-thinned stands as seven "normal stands", which describe seven estates of normal forest. The intention is to study the robustness of carbon storage microeconomics to varying estate characteristics. It was found that the economically optimal rotation ages vary. The state sums of volume and capitalization, corresponding to any optimal rotation, also vary significantly. Growth rates vary more than the optimal expected stand volumes. Consequently, any excess volume related to carbon storage adds on to an almost unified basic volume. For all seven normal estates, the most economical way of increasing carbon storage is to increase the size of trees retained in thinning from above.

**Keywords:** capitalization; capital return rate deficiency; expected value; carbon storage; estate fertility; carbon rent

## 1. Introduction

Boreal forests constitute a significant carbon sink. A particular benefit of boreal regions is carbon storage in the soil. The amount of soil carbon may exceed the carbon storage in living biomass [1–4]. However, living biomass produces the litter that results in soil carbon accumulation, and consequently the rate of carbon storage depends on the rate of biomass production on the site. The biomass production rate is related to the amount of living biomass [2,5–7].

In the occurrence of clearcutting and soil preparation, the net release of carbon from the soil to the atmosphere begins [8–11]. However, recent unpublished datasets indicate that maintenance of canopy cover may or may not be essential in carbon sequestration.

Human activity related to the sequestration and release of carbon into the atmosphere forms a complex system of industries and disciplines [12–16]. Instead of trying to propose any interdisciplinary optimum, we here discuss microeconomics of carbon sequestration in forestry, focusing on the estate level.

In principle, it would be straightforward to apply a carbon trade policy in forestry, possibly administered by public agents [17]. An unbiased carbon trade, however, would incorporate a high initial expense [18]. However, it has been recently shown that a carbon rental system is equivalent to the carbon trade system, without much of an initial expense [18].

From the viewpoint of generic instructions or policy actions, it might be beneficial to reduce the variety of initial estate states by adopting some kind of unifying boundary conditions. A tempting candidate is the normal forest principle [19]. This principle simply refers to the postulation that stand ages are evenly distributed, and stand characteristics are uniquely determined by stand age. Such a postulation, even if often departing from reality, simplifies many treatments, producing idealized systems that are stationary in time. Application of the normal forest principle allows for the determination of an optimal

rotation age, as well as expected values of estate characteristics corresponding to the financial optimum.

Microeconomically optimal rotation age and the expected value of stand volume are not necessarily optimal from the viewpoint of national economics or the viewpoint of carbon sequestration. It has been recently shown that microeconomics often favors solutions with relatively low capitalization [20–23]. Low capitalization reduces volumetric growth rate, as well as litter accumulation rate [21,23,24].

A rather recent paper clarified the microeconomics of carbon sequestration within a boreal spruce forest estate [23]. It found that the best capital return rate is achieved by multiple repeated thinning from above to the transition diameter where sawlogs are gained instead of pulpwood only [21–23]. Increased carbon storage was found to be most economically achieved by increasing the harvesting limit diameter [23]. However, carbon rent derivable from present carbon market prices appeared not high enough to compensate for the induced capital return deficiency [23].

A significant revision to the recent approach [23] has been presented still more recently [25]. Literature values of the yield of sawlogs and pulpwood [26,27] were replaced by empirical observations [25]. The resulting value increment of a typical spruce tree as a function of breast-height diameter is a much smoother function than the literature values would indicate [25]. Some small trees have yielded sawlogs. On the other hand, the expected sawlog content of larger trees is smaller than the literature values would indicate, as operators are trained not to produce sawlogs that may be rejected at the mill [25].

The consequences of smoother value increments are significant [25]. Due to the price premium of sawlogs from clearcutting, all feasible silvicultural treatments on fertile spruce estates terminate in clearcutting instead of multiple repeated thinnings from above [21–23,25]. Financially optimal expected values of timber stocks per hectare are greater when the empirical log yield function is used, in comparison to the literature values of sawlog yield [21,23,25].

The recent treatments have been based on field observations, a growth model, and financial theory. The growth model, as well as the financial treatment, apply probability theory. However, the outcome of the growth model is deterministic, since it is composed in terms of expected values only [21,22,28]. The financial treatment, even if probabilistic as such, applies the normal forest principle, thus applying the postulate of constant stand age distribution within an estate [21–23,25,29,30]. Consequently, any density function of stand age becomes determined by one parameter (rotation age), and the financial treatment becomes deterministic.

The field observations naturally contain statistical scatter. However, recent papers extended the normal forest principle in a somewhat novel manner [23,25]. Instead of discussing the variety of stand properties in detail, a representative "normal stand" was composed as a combination of two stands of average fertility and younger-than-average age. Application of this normal stand procedure has effectively neglected statistical scatter in field observations and rendered the entire treatment deterministic [23,25].

This paper intends to clarify the effects of statistical scatter in field observations. It is of interest whether the results gained when such a scatter is neglected are robust in the presence of the scatter. However, we do not abandon the normal stand - principle. Instead, we applied that principle separately for any of the seven experimental stands where field data has been collected. Each of the seven unthinned stands is used as a normal stand characterizing an estate. The seven normal forest estates is then discussed in order to clarify the robustness of recent results to statistical scatter in field observations. It is worth noting that discussion of any of the seven constituted estates is deterministic—it is of interest whether some of them significantly differ from results gained by neglecting the scatter in field observations.

First, experimental materials are presented, as well as related computational methods. Then, microeconomic methods, as well as carbon rent formulae are introduced. The methods largely unify with a very recent paper [25], apart from the consideration of the scatter

in the characteristics of the normal stands. Third, results are reported for four different kinds of treatment schedules. Finally, the outcome is discussed, and conclusions regarding the robustness of carbon sequestration systems to estate characteristics are given.

## 2. Materials and Methods

### 2.1. Empirical Observations

Eleven circular plots of 314 square meters in area were taken in November 2018 from typical spots of 11 spruce-dominated forest stands in Vihtari, Eastern Finland. The area of the Vihtari estate is 30 ha, with an elevation from 115 to 125 m above sea level. Seven of the stands had experienced only young stand cleaning, whereas four of the stands were previously thinned commercially. Breast-height diameters were recorded as well as tree species, and a quality class was visually determined for any measured tree, reflecting its suitability for adding value in the future.

On measured plots on sites without any previous commercial thinning, the basal area of tree trunks at breast-height varied from 32 to 48 m$^2$/ha, and the stem count from 1655 to 2451 per hectare. On measured plots on stands previously thinned commercially, the basal area varied from 29 to 49 m$^2$/ha, and the stem count from 891 to 955 per hectare. Further details of the experimental stands are reported in an earlier paper [21].

The Vihtari estate was characterized by measurements from the 11 sample plots. However, such a sample did not necessarily represent the entire estate accurately. More importantly, the sampling did not conform to the normal forest principle, with assumptions of constant age distribution and stand characteristics uniquely determined by stand age. Here, we utilize the normal forest principle in terms of establishing a "normal stand" based on the observations and then approximating the development of this "normal stand" as a function of stand age. Introducing an even stand age distribution produces a "normal estate" on the basis of the normal stand, following the normal forest principle [20].

In recent studies [23,25], the normal stand has been established based on two never-thinned sample plots of medium site fertility and a younger range of stand age. The intention was to establish one normal stand representing the typical characteristics of the 11 sample plots measured at the Vihtari estate. The selection of such a representative stand simplified the treatments and rendered the entire analysis deterministic. In other words, it neglected the variation between the 11 sample plots.

The intention of this paper being to investigate the variability among the sample plots, any of the seven never-thinned sample plots is taken as a normal stand. Commercially-thinned stands are neglected since the thinning revenue i not known. Any of the normal stands iss taken to represent a normal estate, representing a "normal forest" [20]. The robustness of recent results to the scatter among sample plots is then investigated in terms of the seven normal estates.

Instead of using literature values [21–23,26] as an estimate of the yield of sawlogs and pulpwood from any tree of particular breast-height diameter, a recently collected dataset of 6123 spruce trees is used [25]. The data was collected by four different single-grip harvesters operated by six individuals at four harvesting sites. Log cutting instructions provided by three different sawmilling companies were applied. One thinning site was located at Vihtari, and two thinning sites and one clearcutting site at Ilomantsi, all in eastern Finland. The area of the sampling territory was 100 km (west to east) by 30 km (south to north). Elevation varied from 115 to 205 m above sea level.

The proportion of sawlogs of the total harvester-measured volume, as a function of breast-height diameter, is shown in Figure 1 [25]. Figure 1 also shows the literature values used in [21–23,26]. It is found that the empirical sawlog yield function is smoother than the function taken from the literature. Some sawlogs were gained from trees of a diameter of 175 mm, unlike in the literature data. On the other hand, the sawlog proportion of trees thicker than 200 mm is smaller in the empirical function. The latter probably is due to harvester operators being trained not to produce sawlogs that would be rejected at the mill.

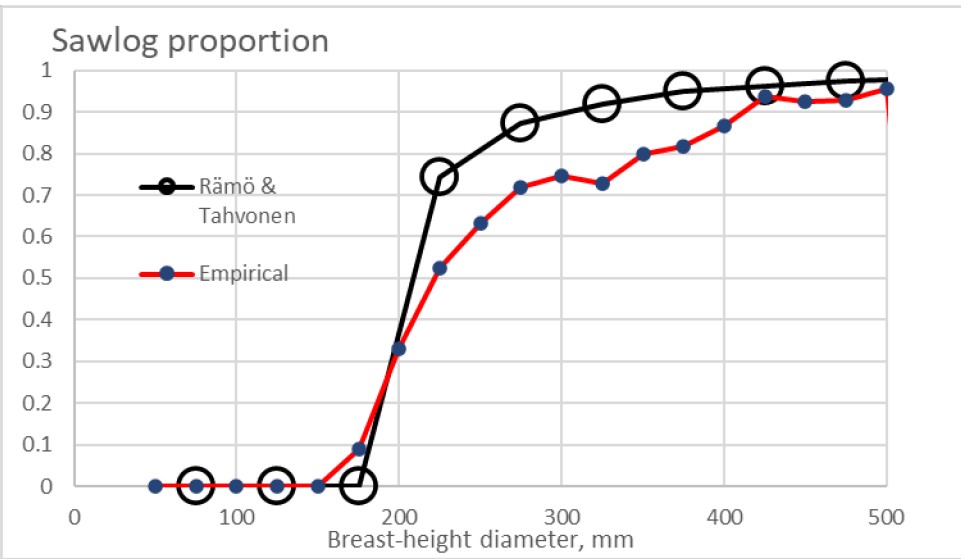

**Figure 1.** Literature values as well as empirical values of the sawlog proportion within the commercial section of spruce stems of different sizes.

### 2.2. Growth Model

For prognostication of further development of the normal stand, some kind of growth model is needed. The growth model of Bollandsås et al. [21,22,28] was adopted, discussing not only growth but also mortality and recruitment. The original growth model [21,22,28] discussed 50 mm breast-height diameter classes within a temporal resolution of five years. Any diameter class was represented by its central tree, and the process of growth was described in terms of the probability of any tree to transfer to the next diameter class [21,22,28]. The underlying idea of the description of growth is that any tree either remains in the same diameter class or transfers to the next diameter class within the five-year time interval. This underlying idea naturally greatly simplifies computation.

It's found from Figure 1 that according to the literature values of sawlog yield, there is a sharp transition between diameter classes centered at 175 and 225 mm. Such a sharp transition would induce a huge value increment. With the empirical yield function being smoother, there is a need for a greater resolution in the tree size description. It is not too complicated to modify the growth model from the size resolution of 50 mm to 25 mm [25]. However, to retain the underlying principle of the growth model, this requires a corresponding change in the temporal resolution, from 5-year time steps to 30 months.

The simultaneous change in the time and size resolutions retains the probability of any tree to transfer to the next diameter class. The same is not true in the case of recruitment and mortality. The latter two quantities are affected by time resolution only. Correspondingly, the recruitment and mortality values become scaled along with the time step by a factor of ½.

For any 25-mm diameter class of trees, the volumetric amount of two assortments, pulpwood and sawlogs are taken from the empirical observations as the expected values for any diameter class.

Description of stand development until the time of observation in 2018 required another kind of approach. An exponential growth function was fitted between an approximated worthless initial volume of 15 m³/ha and the 2018 commercial volume estimate for any of the seven sample plots.

Possibly, the simplest way to approximate financial history is to determine an internal rate of operative return for the period from stand establishment to 2018. In other words, we require

$$V(\tau_3)e^{-s\tau_3} - R(\tau_1)e^{-s\tau_1} - C(\tau_2)e^{-s\tau_2} = 0 \qquad (1)$$

where $R$ is regeneration expense at regeneration time $\tau_1$, $C$ is young stand cleaning expense at cleaning time $\tau_2$, and $V$ is stumpage value of trees at observation time $\tau_3$. In this study, the observation time $\tau_3$ corresponds to November 2018, regeneration time is clarified according to known stand age, and young stand cleaning is assumed to have occurred ten years after regeneration. It is assumed that prices and expenses do not evolve in real terms, and thus presently valid expenses can be used in Equation (1). The regeneration expense is taken as 1250 Euro/ha, and young stand cleaning as 625 Euro/ha.

It is worth noting that the operative internal rate of return $s$ in Equation (1) does not correspond to the capital return rate in the entire activity, as the latter depends on nonoperative capitalization such as bare land value.

### 2.3. Financial Considerations

To determine a momentary capital return rate, we need to discuss the amount of financial resources occupied [21,23,24,31]. This is done in terms of a financial potential function, defined in terms of capitalization per unit area K. The momentary capital return rate becomes

$$r(t) = \frac{d\kappa}{K(t)dt} \tag{2}$$

where $\kappa$ in the numerator considers value growth, operative expenses, interests, and amortizations but neglects investments and withdrawals. In other words, it is the change of capitalization on an economic profit/loss basis. $K$ in the denominator gives capitalization on a balance sheet basis, being directly affected by any investment or withdrawal. It is worth noting that timber sales do not enter the numerator of Equation (2); selling trees at market price levels does not change the amount of wealth, it only converts wealth from trees into the form of cash. However, harvesting naturally changes capitalization appearing in the denominator of Equation (2). Harvesting also likely changes the change rate of capitalization occurring after the harvest.

Equation (2) gives a momentary capital return rate, not necessarily sufficient for management considerations. By definition, the expected value of capitalization per unit area is

$$\langle K \rangle = \int_{-\infty}^{\infty} p(K)K dK \tag{3}$$

where $p(K)$ is the probability density function of capitalization $K$. By change of variables we get

$$\langle K \rangle = \int_0^{\tau} p(K)K\frac{dK}{da}da = \int_0^{\tau} p(a)K(a)da \tag{4}$$

where $a$ is stand age (or time elapsed since latest regeneration harvesting), and $\tau$ is rotation age. The expected value of the change rate of capitalization is

$$\left\langle \frac{d\kappa}{dt} \right\rangle = \int_0^{\tau} p(a)\frac{d\kappa(a)}{dt}da \tag{5}$$

Correspondingly, the expected momentary rate of relative capital return is

$$\langle r(t) \rangle = \frac{\left\langle \frac{d\kappa}{dt} \right\rangle}{\langle K \rangle} = \frac{\int_0^{\tau} p(a)\frac{d\kappa(a,t)}{dt}da}{\int_0^{\tau} p(a)K(a,t)da} = \frac{\int_0^{\tau} p(a)K(a,t)r(a,t)da}{\int_0^{\tau} p(a)K(a,t)da} \tag{6}$$

We find from Equation (6) that the expected value of capital return rate within an estate generally evolves in time as the probability density of stand ages evolves. However, Equation (6) can be simplified to be independent of time by adopting the normal forest

principle, where stand age probability density is constant [20]. Besides, the constancy of the expected value of capital return rate in time requires that prices and expenses do not evolve in real terms. Then, the expected value of the capital return rate becomes

$$\langle r \rangle = \frac{\int_0^\tau \frac{d\kappa(a)}{dt} da}{\int_0^\tau K(a) da} = \frac{\int_0^\tau K(a) r(a) da}{\int_0^\tau K(a) da} \tag{7}$$

It has been recently shown that Equation (7) corresponds to the ratio of the partition functions of the change rate of capitalization and capitalization itself [32]. It also has been recently shown that the maximization of the net present value of future revenues may result in financially devastating consequences [33]. Momentary capital return rate as given in Equation (2) was introduced in 1860 [31]; an expected value was mentioned in 1967 [29,30]; however, applications have been introduced only recently [21–23,25,32,33].

In the context of the improvement harvesting, 20% of the stem count of good-quality trees is removed in all diameter classes due to the establishment of striproads.

The stumpage value is determined in terms of roadside price, deduced by harvesting expense. We here use the roadside prices recently applied by Parkatti et al. [34], 34.04 Euro/m$^3$ for spruce pulpwood, and 58.44 Euro/m$^3$ for sawlogs. We further use the same harvest-expense function as Parkatti et al. [34], stated to be based on a productivity study of Nurminen et al. [35]. In addition to the expense function, we include a fixed harvesting entry expense per hectare. The justification of the latter is that some sites require at least partial pre-harvest cleaning. The entry expense is approximated as 200 Euro/ha. Again, it is assumed that prices and expenses do not evolve in time. In other words, the capital return rate is discussed in real terms.

The original version of the growth model operates in five-year time steps [21–23,28], as discussed above. Consequently, an eventual harvesting entry may take place every five years. According to recent investigations, it often is favorable to harvest every five years [21,23]. However, the fixed harvesting entry cost of 200 Euro/ha restricts low-yield harvesting entries. Numerical investigations indicated that it is not reasonable to harvest if the yield would be less than 20–30 m$^2$/ha. Such an entry limit is in concert with practices applied in the area.

The feasibility of harvesting is investigated in five-year intervals, always considering two options: thinning or clearcut. Clearcutting expenses are lower than thinning harvesting costs, according to Parkatti et al. [34], stated to be based on a productivity study of Nurminen et al. [35]. Moreover, a 15% clearcutting premium for the roadside price of sawlogs is applied, following local tradition. The premium, as well as the harvesting cost reduction, is applied within the last 30-month period before eventual clearcutting.

Thinning procedures are iteratively designed to maximize the expected value of capital return rate, up to the rotation age providing the maximum expected value. After the maximum expected value, any thinning procedure is designed to maximize the capital return rate within the next five years.

Four alternative thinning strategies are investigated for any of the seven normal estates. Two of the strategies are based on thinning from above. The first does not apply any restriction to cutting limit diameters, but the thinning procedure is designed in order to maximize the expected value of capital return rate according to Equation (7). The second does apply a minimal 238 mm diameter limit for thinning from above as a restriction Ref. [25]. The remaining two thinning strategies refer to proportional thinning. In other words, the probability of tree removal is the same in any diameter class. The first of the two proportional thinning strategies is taken as heavy proportional thinning, where 50% of acceptable-quality trees are removed. Gentle proportional thinning removes 30% of acceptable-quality trees in any diameter class.

Application of Equation (7) results in an expected value of capital return rate for any treatment schedule investigated. Treatment schedules not corresponding to the most economical one induce some amount of capital return rate deficiency, in terms of percentage per annum. Such deficiency is related either to the amount of timber stock deviating from the optimal timber stock, or the rotation age deviating from the optimal rotation age, or both. In the former case, the deficiency often is due to an excess standing volume, which allows for expressing the deficiency per excess volume unit.

*2.4. Carbon Rent Considerations*

The last issue in this section of methods regards carbon trade and carbon renting. It is worth noting that carbon prices or rents do not enter the analysis described above in any way. Carbon prices and rents are discussed to enable a comparison of the capital return deficiency per excess volume with any hypothetical carbon rent.

It has been recently shown that policies based on carbon rent are equivalent to policies based on carbon sequestration subsidies and taxies [18]. Unbiased carbon sequestration trade would require a huge initial investment; correspondingly, mostly carbon rent procedures are practically feasible [18]. We here present a brief derivation of the equivalency of the two principles of subsidies, although adopting boundary conditions possibly less restrictive than those of Lintunen et al. [18].

Let us establish a carbon sequestration subsidy system at a particular time $\tau_1$. Within a time range up to time $\tau_2$, the total carbon trade compensation is

$$p_{\tau_1}C_{\tau_1} + \int_{\tau_1}^{\tau_2} \frac{d(pC)}{dt}dt = p_{\tau_1}C_{\tau_1} + \int_{\tau_1}^{\tau_2} \left[ p\frac{dC}{dt} + C\frac{dp}{dt} \right]dt \qquad (8)$$

where $p_t$ is carbon price at time $t$, and $C_t$ is carbon inventory at time $t$. On the other hand, the revenue from carbon rentals is

$$\int_{\tau_1}^{\tau_2} uCdt \qquad (9)$$

where u is the rent rate per carbon unit. Now, to establish equivalency between the carbon storage trade and rent, Equations (8) and (9) must become equal. Equality naturally should apply in any possible circumstance. One of the circumstances is that the time change rate of prices, as well as inventories, is zero. In such a case, the latter term of Equation (8) vanishes. Consequently, the long-term flow of carbon rents should equal a one-time initial storage purchase payment. If the duration of the rent payments extends towards infinity, the only possibility is that the present value of rent payments forms a contracting series. One possibility of such a contracting series is

$$u_{\tau_1}C_{\tau_1}\int_0^\infty e^{-qt}dt = p_{\tau_1}C_{\tau_1} \qquad (10)$$

where $q$ is a discount rate. The corresponding solution for the carbon rent rate is

$$u_t = qp_t \qquad (11)$$

One can readily show that Equation (11) applies not only to a steady-state of Equations (8) and (9) but also to any incremental carbon price and inventory.

## 3. Results

The expected value of capital return rate as a function of rotation age, integrated over the stand lifespan according to Equation (7) is shown in Figure 2. In Figure 2, no restriction is applied in the maximization or the capital return rate. Any feasible treatment cycle terminates in clearcutting, and thus utilizes the premium in clearcutting price, as

well as the harvesting expense reduction. In most cases, thinnings from the above are made to 188 mm cutting diameter limit, even if also 213 mm limits appear. The number of thinnings up to the financially optimal rotation age varies from one to three. After the optimal rotation age, one thinning is triggered in five of the seven cases, in the absence of prior clearcutting.

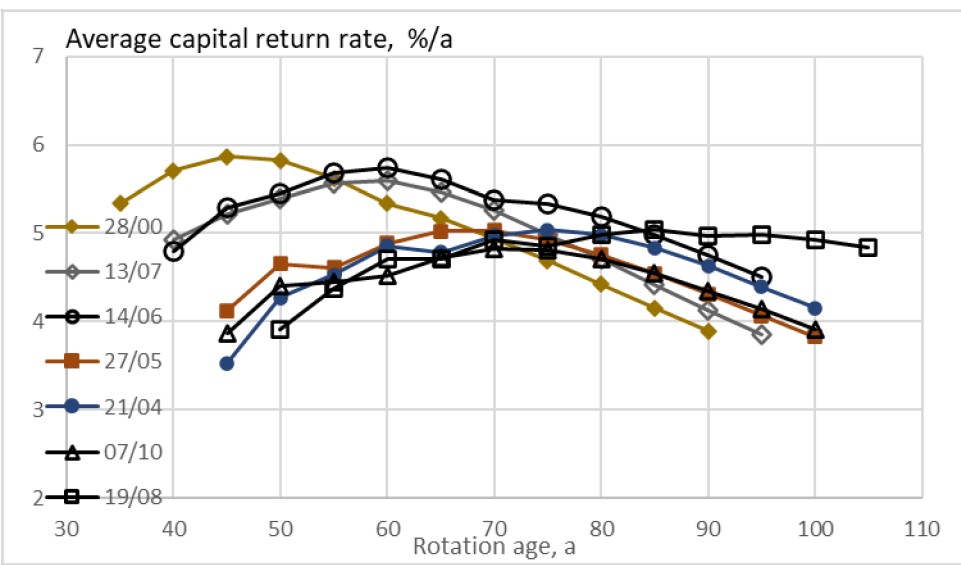

**Figure 2.** Capital return rate according to Equation (7) for seven different normal estates, without any restriction in the design of the applied thinning procedure. The legend identifies the experimental plot identifying any normal estate.

It is found from Figure 2 that the normal estates, with properties determined from the corresponding normal stands, differ in terms of the expected value of capital return rate, but they differed more in the optimal rotation age. The amplitudes of variation are 9.8% and 31%, respectively. The optimal rotation ages span from 45 to 85 years. This is due to fertility, as well as the number and size of trees observed. A third significant contributor is the age of any normal stand in the occurrence of data sampling. The oldest stand was of age 45 years. The capital return rate probably would be higher and the corresponding rotation age lower if the stand had been thinned earlier.

The expected value of capital return rate as a function of rotation age is shown in Figure 3, with the restriction of a minimum cutting limit diameter of 238 mm in thinnings from above. Again, any feasible treatment cycle terminates in clearcutting, and thus utilizes the premium in clearcutting price, as well as the harvesting expense reduction. The restriction induces gentler thinnings and greater capitalizations. The restriction naturally reduces achievable capital return rates. The gentler thinnings reduce the optimal rotation age in five of the seven cases, even if it increases the number of thinnings. The number of thinnings up to the financially optimal rotation age varies from one to six. The optimal rotation ages now span from 40 to 85 years, the latter corresponding to the case of delayed first thinning. The amplitude of variation in capital return rate is 10.2%, and 36% in optimal rotation age.

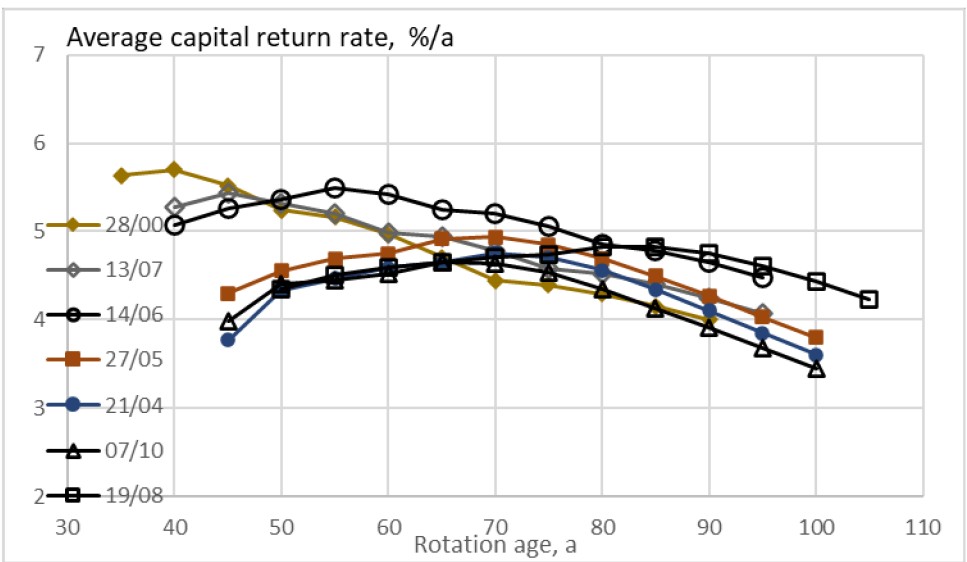

**Figure 3.** Capital return rate according to Equation (7) for seven different normal estates, with a minimum 238 mm cutting diameter limit in thinnings from above. The legend identifies the experimental plot identifying any normal estate.

The expected value of capital return rate as a function of rotation age is shown in Figure 4, with heavy proportional thinning. Again, any feasible treatment cycle terminates in clearcutting and thus utilizes the premium in clearcutting price, as well as the harvesting expense reduction. The heavy proportional thinning renders lower capital return rates in comparison to Figures 2 and 3, as well as shorter rotation times. The optimal rotation ages now vary from 35 to 60 years. The amplitude of variation in capital return rate is 12.9%, and 26.3% in optimal rotation age. In all cases, there is only one thinning. However, in six of the seven cases the financial result is inferior in comparison to no thinning at all.

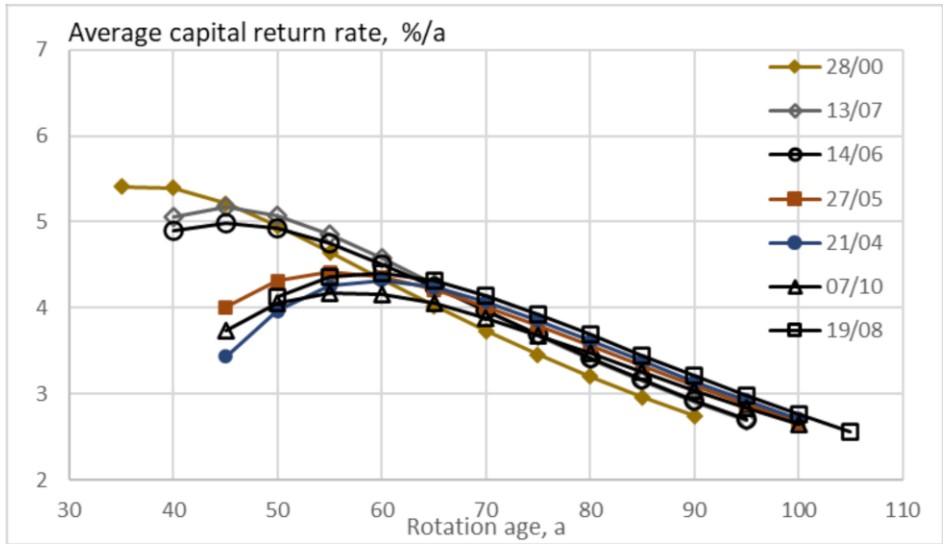

**Figure 4.** Capital return rate according to Equation (7) for seven different normal estates, with heavy proportional thinning. The legend identifies the experimental plot identifying any normal estate.

The expected value of capital return rate as a function of rotation age is shown in Figure 5, with gentle proportional thinning. Any feasible treatment cycle terminates in clearcutting, and thus utilizes the premium in clearcutting price, as well as the harvesting expense reduction. The gentle proportional thinning renders lower capital return rates in comparison to Figures 2 and 3, as well as shorter rotation times. In comparison to the

heavy proportional thinning (Figure 4), capital return rates are higher but optimal rotations shorter. In all cases, there is only one thinning before the rotation age corresponding to the maximum expected capital return rate. Somewhat surprisingly, in five of the seven cases, the financial result is inferior in comparison to no thinning at all. It is, however, worth noting that gaining the clearcutting price premium may be uncertain in the absence of any thinning.

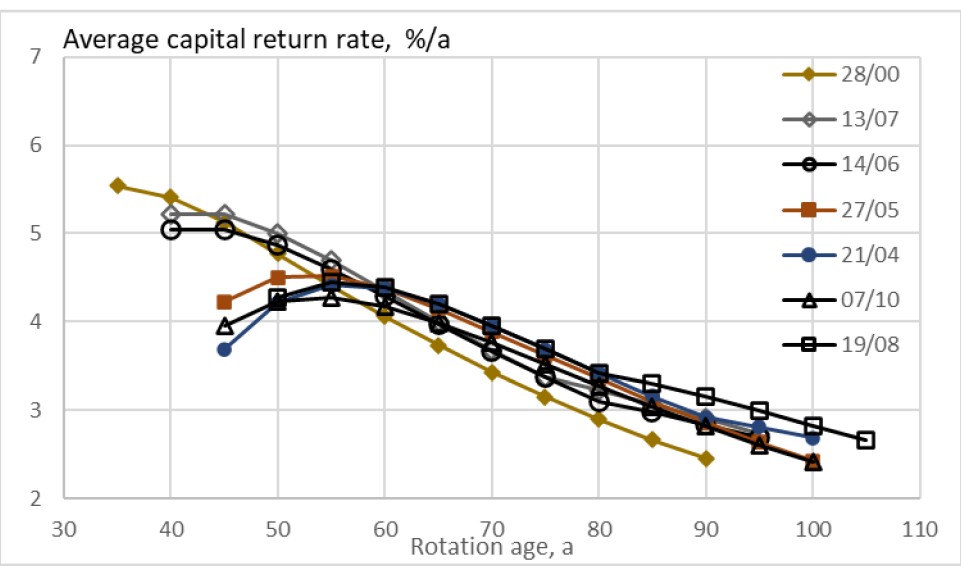

**Figure 5.** Capital return rate according to Equation (7) for seven different normal estates, with gentle proportional thinning. The legend identifies the experimental plot identifying any normal estate.

The capital return rate deficiency in terms of Euros per excess volume is shown in Figure 6, as a function of excess volume, linking economic and environmental sustainability. No restriction is applied in the maximization or the capital return rate. Only positive values are shown in Figure 6. The rotation age providing the maximum capital return rate corresponds to the origin in any series of observations. All the excess volumes in Figure 6 are due to extended rotation age. It is found that even excess volumes of 10 to 25 cubic meters per hectare induced a deficiency of a Euro per excess cubic meter. In the case of the normal stand with retarded thinning, all excess volumes are small.

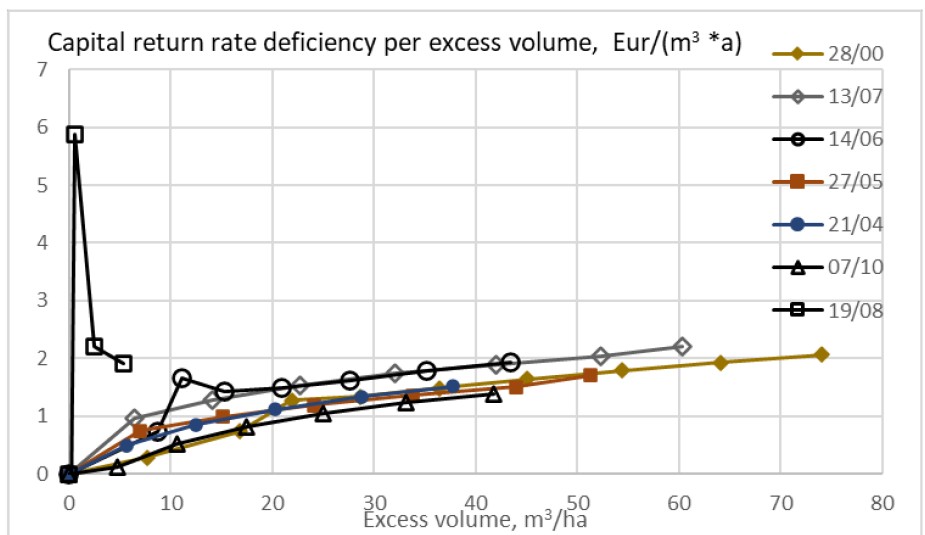

**Figure 6.** Capital return rate deficiency per excess volume unit for seven different normal estates, without any restriction in the design of the applied thinning procedure. Only positive values are plotted.

The capital return rate deficiency in terms of Euros per excess volume is shown in Figure 7, as a function of excess volume. There is a restriction of the minimum cutting limit diameter of 238 mm in thinnings from above. Small excess volumes tend toward large values of the observable, since the excess volume is in the denominator. Moderate excess volumes tend to induce a much smaller capital return deficiency than in Figure 6. It is found that increasing the size of trees remaining after thinning is an economical way of increasing expected stand volume, in comparison to extending the rotations (Figures 6 and 7).

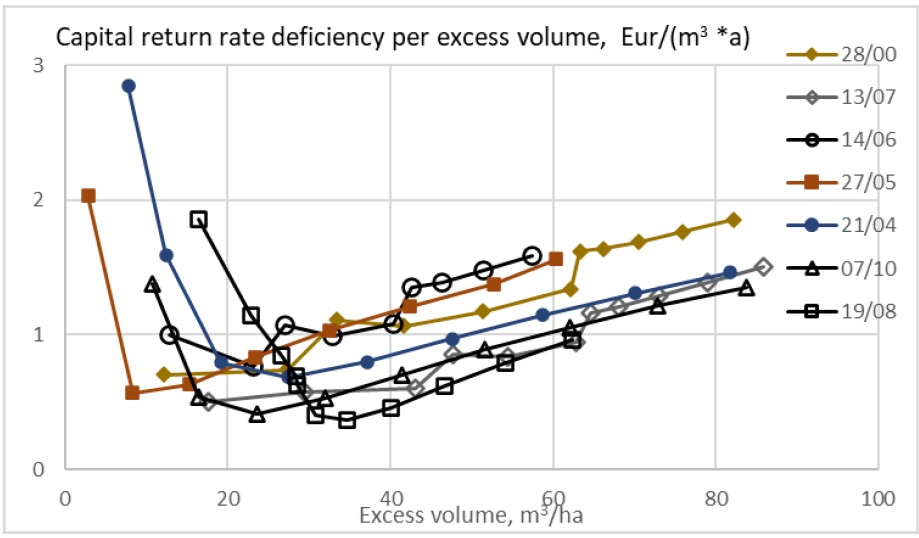

**Figure 7.** Capital return rate deficiency per excess volume unit for seven different normal estates, with minimum 238 mm cutting diameter limit in thinnings from above.

The capital return rate deficiency in terms of Euros per excess volume is shown in Figure 8, as a function of excess volume, for the case of heavy proportional thinning. Again, small excess volumes tend toward large values of the observable, since the excess volume is in the denominator. The capital return rate deficiency appears to be greater than in thinnings from above (Figures 6 and 7), for moderate excess volumes somewhat below two Euros per excess volume unit.

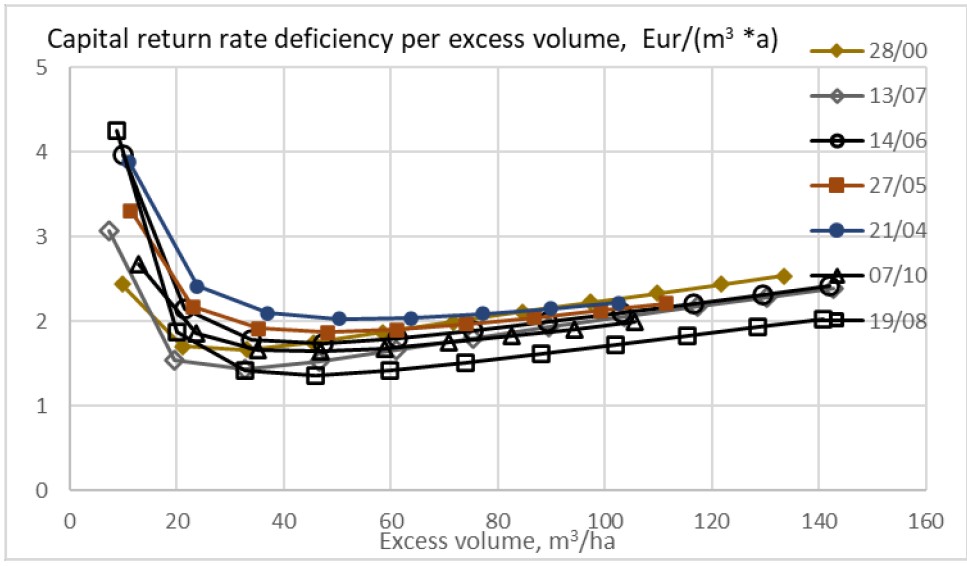

**Figure 8.** Capital return rate deficiency per excess volume unit for seven different normal estates, with heavy proportional thinning.

The capital return rate deficiency in terms of Euros per excess volume is shown in Figure 9, as a function of excess volume, for the case of gentle proportional thinning. The capital return rate deficiency appears smaller than in heavy proportional thinning (Figure 8), but greater than in restricted minimum cutting limit diameter (Figure 7). For moderate excess volumes, the gentle proportional thinning in Figure 9 is competitive to merely extending the rotation age in Figure 6.

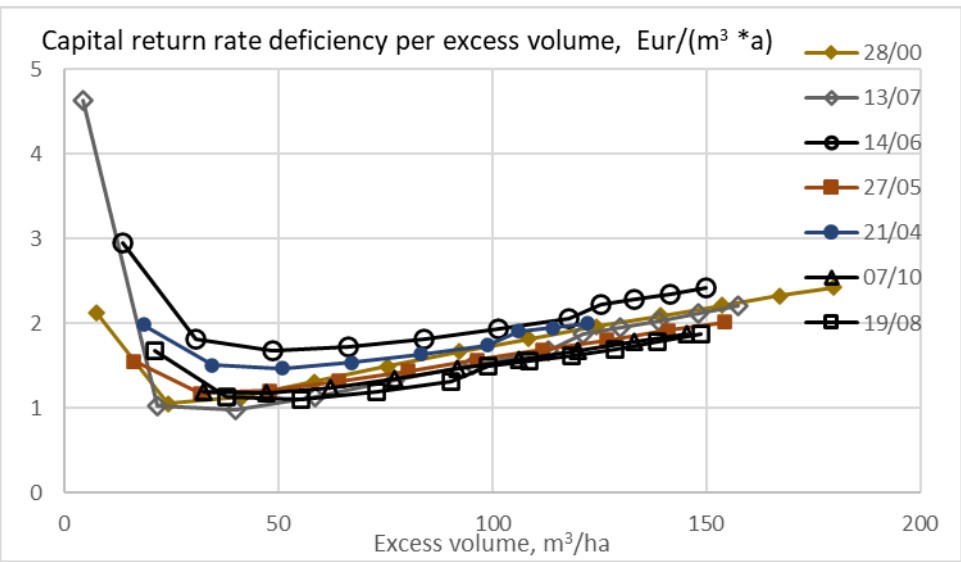

**Figure 9.** Capital return rate deficiency per excess volume unit for seven different normal estates, with gentle proportional thinning.

## 4. Discussion

It appeared from Figure 2 that the optimal rotation age varied significantly. More specifically, the amplitude of variation was 31%. It is of interest to consider whether there are other essential features of the normal stands that would unify. Figure 10 shows the expected value of capital return rate as a function of the state sum of annual volumes per hectare. It is found that at the instant of maximal capital return rate, the state sum of volumes varies significantly, as does the state sum of annual capitalizations (not in the figure). The case with retarded thinning shows more than two times the state sums corresponding to the maximum capital return rate, in comparison to the most fertile stand. The amplitudes of variation of optimal capital return rate and the state sum of annual volumes were 9.8% and 36%, respectively. The amplitude of variation of the optimal state sum of annual capitalizations was 26%.

An obvious reason for the variation in Figure 10 is that sites of different fertility have different time scales in their development. Summing either growing time or some state variables does not unify features unless some time-scaling is applied.

It appears, however, that there is a unifying feature in the seven normal estates. The optimal expected stand volume does not vary much (Figure 11). A consequence is that the excess volumes appearing in Figures 6–9 are additive to an almost unified basic volume, with even the case of retarded thinning differing only slightly. In other words, the amplitude of variation of the optimal expected value of stand volume was 9.1%. This greatly simplifies carbon storage applications.

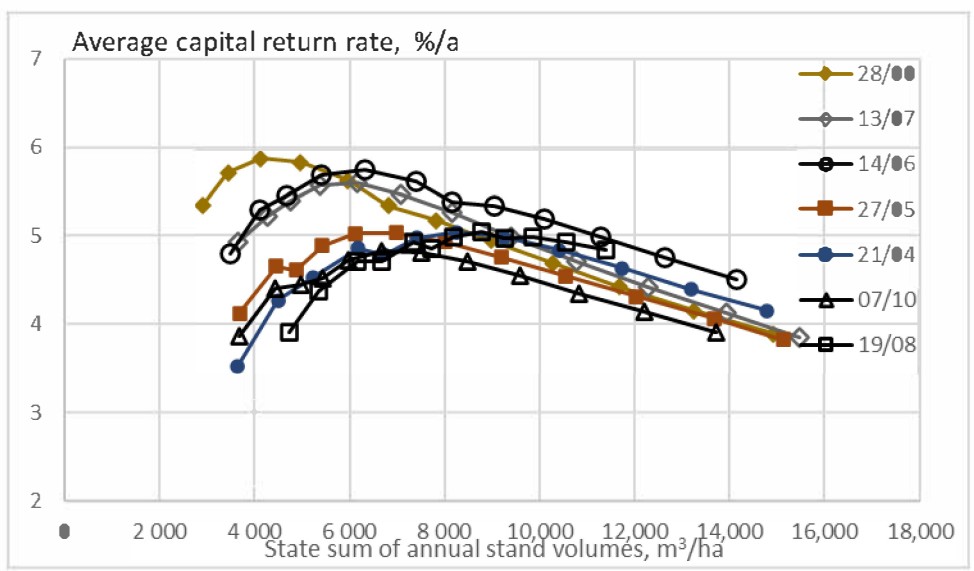

**Figure 10.** Capital return rate according to Equation (7) for seven different normal estates, without any restriction in the design of the applied thinning procedure, as a function of the state sum of annual volumes.

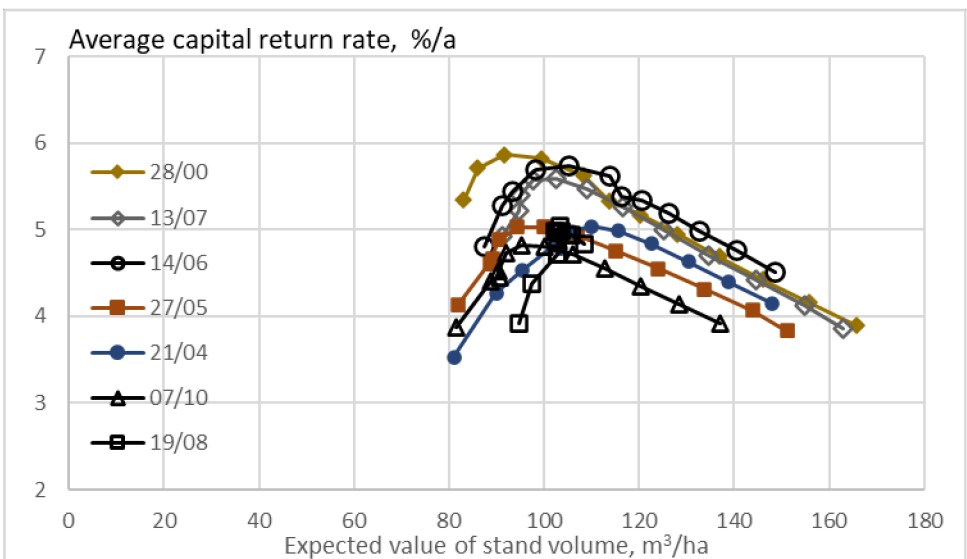

**Figure 11.** Capital return rate according to Equation (7) for seven different normal estates, without any restriction in the design of the applied thinning procedure, as a function of the expected value of commercial volume per hectare.

We have found from Figures 6–9 that for any of the seven normal estates, the most economical way of increasing carbon storage is to increase the size of trees retained in thinning from above. We even find from Figures 7 and 11 that the economics of carbon storage do not vary widely, even if there are different fertilities and growth rates. Correspondingly, the results of the recent investigation [25] appear to be robust to variations among the sample plots collected in the field.

An iterative search for suitable thinning schedules was done up to the rotation age, giving the greatest capital return rate. After the maximum expected value, any thinning procedure was designed to maximize the expected value of the capital return rate observed within the next five years. Any additional five years was thus considered as a marginal extension of the optimal rotation age.

Within the geographic area, professional practice is not to thin aged spruce stands. This is due to a variety of factors, among them avoidance of wind damage and suspected loss of vigor along with aging. The latter might reduce the recovery of trees remaining after thinning. It is of interest to introduce a boundary condition according to which no thinnings are implemented after the optimal rotation age and then compare it with the results appearing in Figures 2, 3, 6 and 7. It is found from Figures 12 and 13 that excess volumes become greater and the increment of capital return rate deficiency smoother in the absence of post-optimum thinnings.

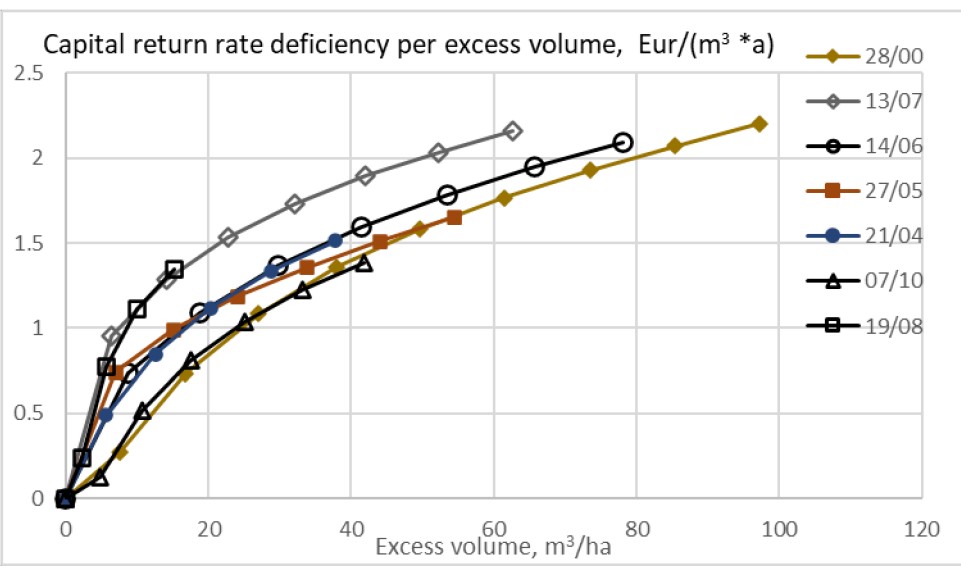

**Figure 12.** Capital return rate deficiency per excess volume unit for seven different normal estates, without any thinning after the financially optimal rotation age.

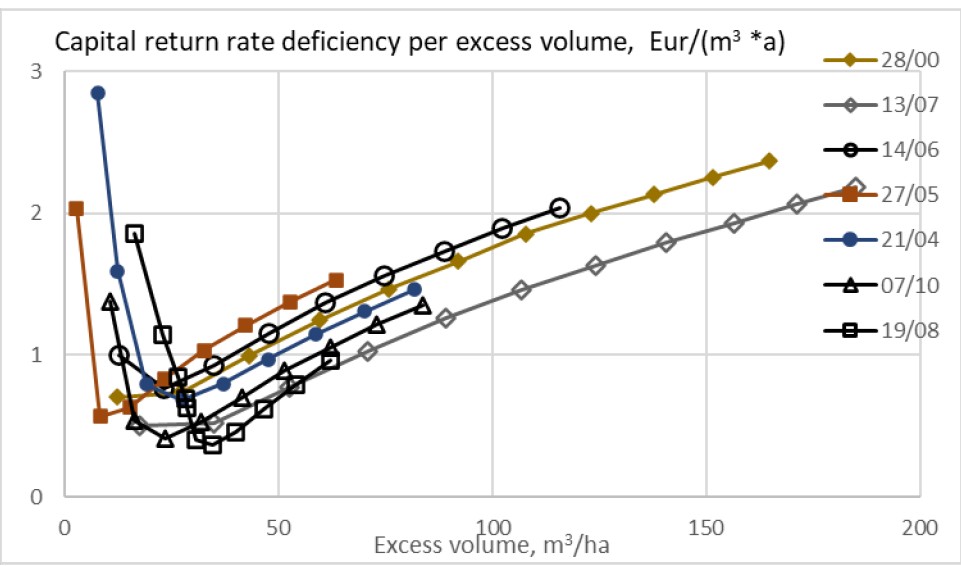

**Figure 13.** Capital return rate deficiency per excess volume unit for seven different normal estates, with minimum 238 mm cutting diameter limit in thinnings from above, and no thinning after the financially optimal rotation age.

The capital return rate being strongly affected by the clearcutting price premium, the five-year marginal approach either does or does not trigger further thinnings. If one would adopt another boundary condition, for example considering a 15-year margin, a greater number of thinnings would be triggered. After the optimal rotation age, observables would

differ. The expected value of the capital return rate would decline more slowly than in Figures 2–5.

If one would maximize the expected value of the capital return rate observed within any 15-year period after the rotation age corresponding to maximal capital return rates, a chart corresponding to Figure 6 would become meaningless. This is because the average stand volume would decrease with time, and the excess volumes would become negative. Instead, setting a restriction in the cutting diameter limit (as in Figure 7) would yield meaningful results. However, with the cutting diameter limit restriction of 238 mm, the excess volumes are still small and partly negative. With 263 mm cutting diameter limit restriction, the excess volumes become positive. The deficiencies are only slightly greater than in Figure 7 and clearly smaller than in Figure 9. A similar result is not achievable by proportional thinning. If one maximizes the expected value of the capital return rate observed within the next 15 years after the rotation age corresponding to maximal capital return rate, the results become inferior in comparison to Figures 8 and 9.

It is further worth noting that multiple thinnings from above would, along with time, transfer the system from the realm of even-aged forestry towards uneven-aged forestry. Such a transition might reduce vigor [36,37], and such a reduced vigor is not considered in the growth model used in this study.

An applicable carbon rent can be derived from carbon storage market price according to Equation (11). At the time of writing, the market price of carbon dioxide emissions is in the order of 25 Euros per ton. This inserted to Equation (11), together with a 3% discount rate [38–40], results as an annual carbon rent of 0.75 Euros per ton of carbon dioxide.

A cubic meter of commercial trunk volume in the boreal forest stores a ton of carbon dioxide (in living biomass, litter, and soil) [1,2,4,7,41,42]. Correspondingly, one might expect a carbon rent of 0.75 Euros per standing cubic meter. In accordance with a recent study [25], that is enough only up to an excess volume of 30 $m^3$/ha. In other words, the recent result is now confirmed considering variation among sample plots measured in the field. For greater levels of carbon storage, the deficiency of capital return rate per standing cubic meter is greater than the appropriate rent.

Any carbon sequestration trade compensation must be proportional to the carbon storage on the estate level [18]. If it would not be, agents with large carbon inventories would initially suffer heavy and unjustified release taxes. Unlike the carbon trade, there is no particular reason why the carbon rent should be proportional. If the carbon rent is made nonproportional, the marginal rent may reasonably reach or exceed the capital return rate deficiency of Figure 7, also on the right of the figure.

It was recently claimed that changing soil fertility as well as changing temperature sum would change the values of all time derivatives appearing in any growth model, as well as in any carbon storage model [23,25]. Correspondingly, capital return rates, as well as annual capital return deficiencies, would change along with all observables resulting from the integration of time derivatives. This statement is confirmed by Figures 2–5 and 10, where both of the axes contain time-dependent quantities. There is a significant scatter among normal estates corresponding to normal stands of different fertility. On the other hand, quantities not involving time derivatives should be robust to fertility. This is found from the horizontal axes of Figures 6–9 and Figures 11–13, which do not contain time derivatives or their integrals. The vertical axis of the latter figures do contain time-dependent quantities, and correspondingly display significant scatter among the normal estates in relative terms. It is, however, worth noting that the variation of optimal capital return rate in Figures 2, 10 and 11 is surprisingly small. The amplitude was 9.8%, in contrast to 9.1% optimal stand volume. The much bigger variations in other time-dependent quantities may be not only fertility-related but possibly also stand age-related.

In addition to soil fertility, there are other aspects of how the seven normal stands differ. Stem counts and stem size distributions differ, as well as stand age at the time of observation in 2018. Such factors do contribute to the characteristics of the normal estates that the normal stands represent. The results above indicate that for the stand of the oldest

age, the first thinning was clearly overdue. Apart from that, the present material does not indicate what would be an optimal stage for the first thinning.

The seven normal stands, representing seven normal estates, were collected from one physical estate in eastern Finland. Correspondingly, the variation among the normal estates reflects variation among stands within the physical estate. The physical estate was completely on mineral soil and relatively productive throughout.

The yield of logs from trees was modeled based on an empirical dataset of 6123 harvested spruce trees. There is no guarantee that such an empirical dataset would apply to all circumstances. The quality of tree trunks varies by stand. Quality requirements of sawlogs vary by sawmill and are subject to change in time. Geographic areas differ, as well as tree species. The yield of logs from trees does differ depending on the type of harvesting (clearcutting, thinning from above/below, continuous cover operations, etc.), stand age, and so on.

Many of the quantitative results in this paper depend on the market prices of round-wood assortments, bare land, silvicultural and harvesting expenses, and carbon emission, as well as the discount rate applied in Equations (10) and (11). Considering eventual changes in the latter two is straightforward in terms of Equations (10) and (11). Changes in the former quantities contribute trivially if they are proportional. In the case of significant nonproportional changes in prices, most of the figures of this paper will have to be redrawn.

## 5. Conclusions

The robustness of recent results regarding microeconomics of carbon storage in fertile boreal forest estates is investigated. For all investigated seven normal estates, carbon stocking can be increased at the expense of a capital return rate deficiency. The most economical way of increasing carbon storage is to increase the size of trees retained in thinning from above, in all seven cases.

Normal stands and normal estates show different characteristics in time-dependent features. However, they unify in time-independent characteristics. In particular, the financially optimal expected value of stand volume per hectare does not vary much, which greatly simplifies carbon storage applications.

**Funding:** This research was partially funded by the Niemi Foundation.

**Institutional Review Board Statement:** Not applicable.

**Informed Consent Statement:** Not applicable.

**Data Availability Statement:** None.

**Acknowledgments:** The Author is obliged to many individuals who contributed to technical arrangements. In particular, the contribution of Jori Uusitalo was invaluable in the analysis of harvester data. Risto-Matti Räsänen, Ari Haapalainen, Keijo Jormanainen, Tero Luukkainen, and Tuure Korhonen kindly assisted in the collection of the yield data.

**Conflicts of Interest:** The author declares no conflict of interest.

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
