# Peer review of "Diversity of Carbon Storage Economics in Fertile Boreal Spruce (Picea Abies) Estates"

_sustainability, doi:10.3390/su13020560_

Round 1
Reviewer 1 Report
I believe the present version is not completely mature for publication. Minor errors, not formatted parts of the ms, too many figures, and a substantially weak conclusion reflect the need of major revisions.
I suggest author to clarify the following points before resubmission.
(i) I encourage author to find a more effective presentation way by incorporating two-four graphs in the same plot or using supplementary materials. It is rather annoying to read few lines of text and then a figure. Comments are relatively short.
(ii) the discussion is absent and conclusions need substantial revision. What is the take home message? What is the linkage between economic and environmental sustainability in this paper? Are estates studied here representative of what?
(iii) literature review should be extended a bit, giving more emphasis to non-European studies, e.g. Canada.
(iv) formats should be carefully checked in some points (yellow highlights: why?). Minor language errors throughout the paper. Be careful!
Author Response
Thank you very much.
Yes, indeed, busy time schedule ahead of the special issue deadline obviously has contributed to the finalization of the manuscript.
The Author agrees that there were too many Figures. All Figures have been redrawn, and two Figures have been removed. The issues covered in the removed Figures (in the Discussion) are now explained in text only.
The Reviewer states that the conclusions are substantially weak. The author disagrees. For an empirical study within a practical discipline, the conclusions presented appear exceptionally strong.
There are two conclusions. First, for all seven normal estates, the same rules for carbon storage economics apply. These rules further are the same as those reported in a recent study for one single normal forest. This strong result is obviously fortunate: reality could be a mess, but it is not. Technically, the most economical way of increasing carbon storage is to increase the size of trees retained in thinning from above.
The second significant result confirms statements previously presented on the basis of theoretical/dimensional arguments. It was stated in Ref. [25]:
” Changing soil fertility, as well as changing temperature sum, would change the values of all time derivatives. Correspondingly, capital return rates, as well as annual capital return deficiencies would change. Greater fertility would result in greater time change rates and vice versa.”
In the Discussion of this paper, it is stated:” …quantities not involving time derivatives are supposed to be robust to fertility. This is found from the horizontal axis of Figs. 6, 7, 8, 9, 12, and 13, that do not contain time derivatives or their integrals. The vertical axis of the latter Figures do contain time-dependent quantities...”
For a pure theorist, experimental verification of theoretical arguments might not be important. For most of us, it is. Again, the reality could be a mess, but fortunately, it is not.
A peculiarity is the small variation in the achievable capital return rates. This is now discussed in the Discussion.
Discussion related to the variability of observables has been rewritten, in order to improve clarity.
The Discussion has not been “absent”. In the original version, Figs. 2 to 9 expressed results. Figs. 10 to 13 were in the section “Discussion”. The subtitle was in place. By definition, “Discussion” contains ponderings related to reliability and consistency of the Results, as well as their implications. Why were the Figures in the Discussion? Because they were needed to discuss the Results of the previous section.
The discussion has been rewritten. In particular, statements clarifying what the seven normal stands represent, and what they do not represent, has been amended. The Discussion now focuses on boundary conditions that more closely adhere to conventional professional practices within the study area. This new discussion is accompanied by two previously nonexistent Figures. Correspondingly, the total number of Figures finally is the same as in the original manuscript.
As mentioned above, there are two conclusions. The first paragraph of the section “Conclusions” states the first conclusion. The second paragraph states the second conclusion.
The conclusions mentioned above reflect the objective of the present study. There are other significant issues a novel reader may “take home”. They are in Figs. 6 to 9, as well as 12 and 13. These six Figures show capital return rate deficiency per excess volume as a function of excess volume, linking economic and environmental sustainability. The link is now explicitly mentioned in the text. There are six different Figures since there are six different sets of boundary conditions investigated. Four of them appear in the Results. The remaining two are variants of the earlier ones, appearing in the Discussion.
North American carbon sequestration studies have been referred to. However, the Author is not aware of any Non-Eurasian investigations regarding capital return rate deficiency.
After all the rewritings, a careful linguistic revision has been implemented. The Author is not aware of the origin of the yellow highlights, but it appears that the Editorial office of the Journal has found a way to get rid of them.
Reviewer 2 Report
The author proposes a model to quantify carbon storage in a boreal forest. The paper is interesting and adds advances to science knowledge, giving the importance of the problem. The methods are sounds and described with rigour. The results are in line with the expectations and commented with competence. I only suggest a careful check of the text, since there are many typos, and a better presentation of figures, many of which are not clear.
Some other minor suggestions are reported in the commented text in attachment.

Author Response
Thank you very much.
All the Figures have been redrawn, related to some technical editions in the underlying spreadsheets. Also, the presentation of the Results has been redesigned, related to the application of different kinds of boundary conditions. In particular, the Discussion now focuses on boundary conditions that more closely adhere to conventional professional practices within the study area.
The author does not think specific values of variability belong to the Abstract. However, they have now been introduced in the Results, and partially also in Discussion.
The structure of the paper not being completely trivial, the Author fees it is necessary to outline the remaining part of the paper at the end of the Introduction.
Wikipedia:” The abbreviation cf. (short for the Latin: confer/conferatur, both meaning "compare") is used in writing to refer the reader to other material to make a comparison with the topic being discussed.”
In front of the statement “The regeneration expense is taken as 1250 Eur/ha, and young stand cleaning 625 Eur/ha.” there is another sentence “It is assumed that prices and expenses do not evolve in real terms, and thus presently valid expenses can be used in Equation (1).”
The sentence appearing first is supposed to indicate that the values mentioned are taken from the current expenses in the area.
Reviewer 3 Report
The authors of the paper Diversity of Carbon Storage Economics in Fertile Boreal Spruce (Picea Abies) Estates present a relevant topic both by the fact that "a" normal forest ", an idealized estate, with a uniform distribution of stand ages, can be used in the study of practices sustainable management ”, but especially as it achieves the priority of global climate change.
Concepts, bibliographic sources and citations are appropriately mentioned in the paper. For example, the amount of carbon in the soil can exceed the storage of carbon in living biomass [1-4]. The research methodology is appropriate, the authors of the research use research tools based on empirical data and models such as the Growth Model of Bollandsås et al. [21,22,28], Original growth model [21,22,28] and Growth model from the size resolution of 50 mm to 25 mm [25].
The research results are presented by the research authors having presented results such as: the expected value of the rate of return on capital depending on the state amount of annual volumes per hectare and Fig. 11 depending on the state amount of annual capitalizations. Moreover, the authors present the fact that “the average volume of the stand would decrease in time, and the excess volumes would become negative. Instead, setting a restriction on the cutting diameter limit (as in Fig. 7) would produce significant results. However, with the cut-off diameter limit of 238 mm, the excess volumes are still small and partially negative. However, in order to clearly highlight the scientific contributions of the authors to the literature, we suggest that the authors of the research mention in a paragraph what these personal results are.
The conclusions are presented in a limited way, especially in the context of the fact that the topic of the paper is very relevant for the global priorities. Therefore, we suggest to the authors the completion of the conclusions with the personal scientific elements resulting from the paper, from the limitations identified in the research, as well as from the future researches in the continuity of the studied topic. We congratulate the research team for the topic studied, and after reviewing the above mentioned, we propose for acceptance the paper.
Author Response
Thank you very much.
Yes, the results should have been presented more clearly. Also, the limitations and restrictions should have been discussed more thoroughly, and the conclusions should have been clearer.
All the Figures have been redrawn, related to some technical editions in the underlying spreadsheets. Also, the presentation of the Results has been redesigned, related to the application of different kinds of boundary conditions. In particular, the Discussion now focuses on boundary conditions that more closely adhere to conventional professional practices within the study area.
Many of the statements the Reviewer quotes in the second-last paragraph would be at least confusing, if not impossible to understand. However, a detailed examination revealed that such statements do not appear in the paper. There is nothing like “state amount” in the paper. Instead, there is a “state sum”. It is a common concept in probability theory. In physics, a concept “partition function” is often used. However, in the mind of the author, “state sum” better fits here. Correspondingly, the figure caption “Capital return rate according to Eq. (7) for seven different normal estates, without any restriction in the design of the applied thinning procedure, as a function of the state sum of annual volumes.” appears to be correct, and exactly describes the content of the Figure. The author does recognize that state sums are not customarily discussed in forestry (however confer to Ref. 32). However, “Sustainability” is not a forestry journal.
The same applies to the second inclarity example provided by the Reviewer. It was stated in the paper: “If one would, after the rotation age corresponding to maximal capital return rate, maximize the expected value of the capital return rate observed within the next 15 years, a chart corresponding to Fig. 6 would become meaningless. This is because the average stand volume would decrease with time, and the excess volumes would become negative.” The statement is completely correct. An excess volume can be negative. However, from the viewpoint of carbon storage applications, negative excess volumes do not make any sense. This is because there is a capital return rate deficiency (negative implication) that is connected to reduced carbon storage (negative cause).
It is worth noting that the sentences quoted above are not in the “Results” -section of the paper. They are in “Discussion”. The statement in the Discussion “… Instead, setting a restriction in the cutting diameter limit (like in Fig. 7) would yield meaningful results. However, with the cutting diameter limit restriction of 238 mm, the excess volumes are still small, and partly negative. ..” The text appears correct.
There are important issues in the last paragraph of the Reviewer. However, the Author thinks the discussion of study limitations, as well as future prospects, belongs to “Discussion” instead of “Conclusions”. The section “Discussion” has now been amended in these respects.
Round 2
Reviewer 1 Report
The revision work was carried out in an appropriate way. THank you.